# MitoQ Is Able to Modulate Apoptosis and Inflammation

**DOI:** 10.3390/ijms22094753

**Published:** 2021-04-30

**Authors:** Elisa Piscianz, Alessandra Tesser, Erika Rimondi, Elisabetta Melloni, Claudio Celeghini, Annalisa Marcuzzi

**Affiliations:** 1Hygiene and Public Health Unit (ASUGI), 34129 Trieste, Italy; elisa.piscianz@asugi.sanita.fvg.it; 2Department of Pediatrics, Institute for Maternal and Child Health—IRCCS Burlo Garofolo, 34137 Trieste, Italy; alessandra.tesser@burlo.trieste.it; 3Department of Translational Medicine and LTTA Centre, University of Ferrara, 44121 Ferrara, Italy; elisabetta.melloni@unife.it; 4Department of Translational Medicine, University of Ferrara, 44121 Ferrara, Italy; claudio.celeghini@unife.it (C.C.); annalisa.marcuzzi@unife.it (A.M.)

**Keywords:** inflammation, autophagy, mitochondria, cytokines, cholesterol

## Abstract

Mitoquinone (MitoQ) is a mitochondrial reactive oxygen species scavenger that is characterized by high bioavailability. Prior studies have demonstrated its neuroprotective potential. Indeed, the release of reactive oxygen species due to damage to mitochondrial components plays a pivotal role in the pathogenesis of several neurodegenerative diseases. The present study aimed to examine the impact of the inflammation platform activation on the neuronal cell line (DAOY) treated with specific inflammatory stimuli and whether MitoQ addition can modulate these deregulations. DAOY cells were pre-treated with MitoQ and then stimulated by a blockade of the cholesterol pathway, also called mevalonate pathway, using a statin, mimicking cholesterol deregulation, a common parameter present in some neurodegenerative and autoinflammatory diseases. To verify the role played by MitoQ, we examined the expression of genes involved in the inflammation mechanism and the mitochondrial activity at different time points. In this experimental design, MitoQ showed a protective effect against the blockade of the mevalonate pathway in a short period (12 h) but did not persist for a long time (24 and 48 h). The results obtained highlight the anti-inflammatory properties of MitoQ and open the question about its application as an effective adjuvant for the treatment of the autoinflammatory disease characterized by a cholesterol deregulation pathway that involves mitochondrial homeostasis.

## 1. Introduction

The mitochondria represent the main sources of reactive oxygen species (ROS) related to several diseases caused by alteration of cholesterol biosynthesis [1,2,3]. In the past years, new drugs belonging to the Mitochondria-Targeted Antioxidants (MTAs) family, such as Mitoquinone (MitoQ), Mitotempo and MitoVitE, have proven effective in decreasing oxidative stress across multiple disease models. MTAs are indeed known for their activity against the activation of the mitochondrial damage because of their antioxidant and mitochondrial protective properties. In particular, MitoQ is characterized by high bioavailability, and several prior studies have demonstrated its neuroprotective potential [4,5].

Of note, although the FDA (U.S. Food and Drug Administration) has not yet approved MitoQ as a drug, 19 studies are currently registered to evaluate the effect of MitoQ in several diseases, such as multiple sclerosis, Parkinson’s disease, Alzheimer’s disease and cystic fibrosis [6].

Cholesterol in cells is essential for the correct organization of the cell membrane, determines its permeability and fluidity and is considered an important factor that regulates the plasticity of membranes [7]. This is especially important for mitochondria, which are considered the energy source of the cell, in which cholesterol is fundamental to membrane maintenance [8,9,10]. Moreover, the cholesterol homeostasis in the mitochondria is necessary to ensure and maintain the mitochondrial biogenesis [11]. Several works have provided evidence about the regulation of microtubule-based mitochondrial trafficking, highlighting that mitochondrial motility affects neuronal growth and synapses [12,13]. Consequently, the mitochondrial dynamics are particularly sensitive to various stimuli and the mechanism underlying its regulation is a balance between mitochondrial fusion and fission [14]. The alterations or perturbations in mitochondrial dynamics can been observed in several neurodegenerative diseases [15,16].

These dysregulations are more easily detectable in the central nervous system, since this area represents a system closed to cholesterol synthesis [17]. It is well known that statins, such as lovastatin, are able to induce apoptosis in neuronal cell lines and that the mevalonate prevents the programmed cell death induced by the blockade of this pathway [18,19].

In previous studies, indeed, we demonstrated that in neuronal cells the blockade of the cholesterol pathway, also called mevalonate pathway, using a statin, induced significant morphological changes and dysfunctions in mitochondria [20] and that the presence of antioxidant MitoQ plays a fundamental role as preventive agent [17]. 

To date, the molecular mechanism undergoing the MitoQ role in this experimental cellular model is not known and this is an important step to improve our knowledge for several pathogenesis linked to cholesterol metabolism (Figure 1).

It has been shown that blocking of the pathway of cholesterol metabolism is associated with increased susceptibility to pro-inflammatory stimuli which induces the activation of inflammation. 

Important regulators of the inflammatory response are represented by the inflammasomes, cytosolic multiprotein complexes that mediates the activation of the pro-inflammatory caspase 1 [21,22]. These enzymes are key regulators of the conversion of the inactive pro-IL-1β into the mature active IL-1β and promote pyroptosis, a form of programmed cell death. The most investigated inflammasome is NLRP3 (NACHT, LRR and PYD domains-containing protein 3), which has been associated with various diseases considering that caspase 1 activation is crucial for starting inflammation in response to injury or infection [23]. In general, one of the most important regulator of inflammasomes is represented by autophagy, a complex mechanism that maintains cellular homeostasis by the degradation of unnecessary or dysfunctional components through the action of lysosomes. Several studies have established a strict interplay between the inflammasome and autophagy, and the balance between these systems is crucial for various biological functions, such as inflammation, metabolism, programmed cell death and cell differentiation [24,25,26]. 

The present study aimed to identify the components of the inflammation platform and examine the impact of this inflammatory system on the mitochondria functions, associated with the blockade of mevalonate pathway, in a human-derived desmoplastic cerebellar medulloblastoma cell line used as a neuronal model. In particular, the focus of this study was to establish the role of the MitoQ supplementation on the mitochondria metabolism and the timepoints of this action. The understanding of this system may provide insights into disease pathogenesis that might serve as potential targets for therapeutic intervention.

## 2. Results

### 2.1. Effect of MitoQ on Mitochondrial Electrochemical Potential Gradient

To set the experimental conditions and verify the mitochondrial damages triggered from the blockade of metabolic pathway, we employed a cytofluorimetric assay, using the lipophilic membrane-permeant JC-1 dye that accumulates in mitochondria and detects mitochondrial depolarization occurring in a variety of conditions, including apoptosis [27,28]. JC-1 staining confirmed that the statin caused significant damage to mitochondrial membrane after an extended incubation (48 h, Lova: 73.45 ± 14.61 vs. untreated: 24.96 ± 9.41; *** *p* < 0.0001), not so evident in the short period (12 h, Lova: 27.87 ± 0.81 vs. untreated: 17.17 ± 2.25; * *p* < 0.05) (24 h, Lova: 35.38 ± 3.52 vs. untreated: 23.20 ± 4.28; * *p* < 0.05). The presence of MitoQ limited apoptosis of DAOY cells in the early-stage analysis (12 h, MitoQ + Lova: 14.08 ± 1.05 vs. Lova: 28.30 ± 0.42; §§ *p* < 0.01) and not at 24 and 48 h (Figure 2). Moreover, the data show that MitoQ alone is comparable to untreated condition.

### 2.2. Inflammatory Genes Expression Related to MitoQ Activity

To investigate the activation of inflammatory platforms linked to apoptotic systems, we measured the expression levels of NLRP3, CASP-1 (caspase 1) and OPA-1 genes by qPCR in our experimental cellular model. Figure 3 shows that lovastatin treatment for 12 and 24 h induced a hyper-expression of NLRP3, CASP-1 and OPA-1 compared to the untreated condition. Although less evident, the same trend is detectable after 48 h of lovastatin challenge only for OPA-1 gene, but not for NLRP3 and CASP-1 genes. 

The 12 h combined treatment of lovastatin and MitoQ reduced NLRP3, CASP-1 and OPA-1 gene expression levels in comparison with lovastatin alone; the downward trend was maintained for NLRP3 even if it was less evident for CASP-1 and OPA-1 at the 24 h challenge. These results are not confirmed after 48 h of the combined treatment: NLRP3 and CASP-1 expression levels can be compared with the condition of lovastatin alone, while the decreasing trend of OPA-1 expression is slightly appreciable. 

Taken together, these results confirm that the addition of MitoQ to lovastatin is able to limit the inflammation platform in our cellular model in a short treatment timeframe (12 and 24 h). Moreover, we can assume that 48 h is a too long stimulation timepoint for our experimental design.

### 2.3. Pro-Inflammatory Cytokines Secretion

The cytokines profile at the different timepoints showed a statistically significant increase production of all analytes (IL-1β, IL-2, IL-4, IL-6, IL-8, IL-17, IFN-γ and TNF-α) at 12 h after lovastatin treatment in comparison to untreated condition. Furthermore, the presence of MitoQ was able to reduce the secretion of all cytokines and in particular the decrease was appreciable for IL-1β, IL-2, IL-4, IFN-γ and TNF-α (Table 1).

At 24 and 48 h after lovastatin treatment, alone or with MitoQ, the cytokines secretion did not show a significant modulation (Appendix A).

### 2.4. Influence of MitoQ on Mitochondrial Complexes in DAOY Cell Line Treated with Lovastatin 

We used the Luminex technology to obtain the relative expression of the mitochondrial respiratory Complexes I–V proteins to assess mitochondrial activity in our experimental cellular model.

Measurement of the mitochondrial complexes (I–V), 12 h after the treatment with lovastatin, revealed a significant increased activity and these results persist at 24 and 48 h after treatment just on Complexes IV and V (Figure 4). Of note, MitoQ was not able to counteract the lovastatin action, except for Complex V. As shown in Figure 4, after 48 h of treatment with MitoQ and lovastatin, we could evidence a loss of activity of the Complex V compared to lovastatin alone. MitoQ, indeed, in this condition, showed a strong inhibitory effect, and this result indicates a decreased mitochondrial activity during Lova + MitoQ stimulation.

## 3. Discussion

The cholesterol in the brain represents approximately 25% of the total cholesterol present in our body, containing about 10-fold more than any organ [29]. In the central nervous system, high cholesterol levels are essential in the myelin sheath to facilitate the transmission of electric signals and it is also necessary for synapse and dendrite formation. Furthermore, this sterol is fundamental for neuronal physiology, both during development and in the adult stage. Notably, several disorders are associated with defects in cholesterol synthesis and metabolism; there are, indeed, a large family of diseases caused by inborn errors of metabolism and including rare pathologies such as Niemann–Pick C disease, mevalonate kinase deficiency and Smith–Lemli–Optiz syndrome [30,31]. In particular, most of the pathogenesis of these diseases suggest that there is a common involvement of the inflammation platform [32]. The NLRP3 inflammasome represents a physiological trigger to activate the inflammation, as confirmed by the results obtained after 12 and 24 h of treatment with lovastatin. It is therefore not surprising that in this study caspase 1 (CASP-1) showed a comparable activation trend at 12 h, and that the addition of MitoQ was able to modulate and decrease this expression due to the blockade of the pathway. These data were statistically significant 12 h after the stimulus: MitoQ, indeed, was able to prevent or limit the inflammation activity, but this ability decreased at later timepoints (24 and 48 h). 

The total absence of effects in long-time treatments could be explained by the MitoQ half-life or by the severe cellular morphological changes and damage due to lovastatin, a condition in which MitoQ action is not effective. The homogeneous cytokines profile at 12 h confirms that at this timepoint the MitoQ was able to counteract the inflammatory activation supported by gene expression. Data obtained with the membrane-permeant JC-1 dye, used to monitor mitochondrial health, confirm that the mitochondrial membrane was damaged after 48 h of lovastatin, regardless of MitoQ treatment [33]. In contrast, at 12 h, we can observe that MitoQ is able to counteract the lovastatin treatment, supporting the hypothesis that MitoQ can modulate the morphologic change in a short time.

It is important, instead, to highlight that the high significant OPA-1 expression level after 12 h of lovastatin treatment is strictly associated with its biological role in the regulation of mitochondrial stability and cellular energy output, as demonstrated in previous studies [34,35]. Indeed, OPA-1 is a mitochondrial fusion protein which is characterized by rigorous cellular regulation, including apoptosis and respiratory capacity, necessary for oxidative phosphorylation (OXPHOS) and ATP production [36,37,38].

The obtained data are notably relevant in all time analyzed since lovastatin treatment in OXPHOS-Complex V induced a significant increase in comparison with untreated condition. It should be noted that MitoQ at 48 h was able to reduce the Complex V activity: these results can be interpreted as a physiological mitochondrial mechanism to protect the membranes. Notoriously, OXPHOS-Complex V, also called ATP synthase, is the final enzyme in the oxidative phosphorylation pathway and represents the energy source of the system. This evidence suggests that the ATP synthase is correlated with the fusion of the mitochondrial inner membrane regulated by OPA-1 [35]. The fundamental role of this protein in mitochondria stability and in the energy production process is attested by its deficiency in the most common age-related neurodegenerative diseases such as Alzheimer’s diseases and Parkinson’s disease [34,35,39,40]

Moreover, literature data demonstrate that OXPHOS activity was correlated to NLRP3 inflammasome activation in several diseases [41,42,43,44] so much that NLRP3 may be considered as a novel biomarker and therapeutic target for these pathologies [45].

It is interesting to highlight that just NLRP3 expression remains at 24 h, and these data were confirmed by IL-1α secretion, validating the timepoints that interplay between these phenomena and supporting the role of inflammasome in neurodegenerative diseases [23,46]. The same trend is found with IL-6 and TNF-α, which together with IL-1β are considered as key players of the neuroinflammation, since they are the interface between the immune system and the nervous system. These cytokines, indeed, are released in response to multiple stimuli able to activate the immune response and interact with the activity of glial and neuronal cells, modulating for example the sensitivity of synapses to the activation of membrane receptors for glutamate, GABA or endocannabinoids and regulating the release of neurotransmitters such as dopamine [47,48]. Moreover, it is interesting to highlight the modulation of IFN-γ, known to promote the expression of TNF-α and IL-1 β in microglia, which suggests a key role of this cytokine in the development of immune and inflammatory responses involved in neurodegeneration mechanisms in the CNS [49,50,51]. Moreover, previous studies have revealed IL-17 as a cytokine involved in chronic inflammatory neurological diseases, and we can confirm that, in our neuronal model, it can be modulated from the inflammation [52,53]. All these data are relevant for the proper understanding of cytokines signaling pathways involved in the regulation of neuroinflammation and could be crucial for the development of new therapeutical strategies [54].

## 4. Materials and Methods

### 4.1. Reagents and Cell Culture

Lovastatin (Lova, Mevinolin from Aspergillus terreus) (Sigma-Aldrich, Milan, Italy) was resuspended in ethanol (Sigma-Aldrich, Milan, Italy) not exceeding 0.01% final concentration per well. MitoQ, kindly provided by MP Murphy (MRC Mitochondrial Biology Unit, Cambridge, UK), was resuspended in ethanol (Sigma-Aldrich, Milan, Italy) [6]. DAOY cell line was purchased from ATCC (ATCC^®^ HTB-186™) and cultured in EMEM (Eagle’s Minimum Essential Medium, Euroclone, Pero, Italy), supplemented with 10% foetal bovine serum (FBS, Euroclone, Pero, Italy), 2 mM glutamine and 100 U/mL penicillin/streptomycin. Twenty-four hours after seeding, cells were pre-treated with MitoQ (200 nM) for 1 h and then treated with Lova (10 μM) [17].

### 4.2. JC-1 Assay

The lipophilic cation dye JC-1 (5′,6,6′-tetrachloro-1,1′,3,3′-tetraethylbenzimi-dazolylcarbocyanine iodide) assay kit was purchased from Cayman Chemical Company (Ann Arbor, MI, USA), and the analyses of the mitochondrial membrane potential were performed strictly according to the manufacturer’s protocol.

Fluorescence was acquired with FACSCalibur cell analyzer and CellQuest software, version 5.1.1 (Becton Dickinson, Franklin Lakes, NJ, USA) and then analyzed with FlowJo software (version 7.6, Treestar, Inc., Woodburn, OR, USA).

### 4.3. RNA Isolation, Reverse Transcription and Real Time-PCR (qPCR)

Total RNA was extracted from cells with TRIzol reagent (Thermo Fisher, Waltham, MA, USA) and its quality was estimated using 2100 Bioanalyzer System (Agilent Technologies, Santa Clara, CA, USA). RNA was reverse transcribed into cDNA with High Capacity cDNA Reverse Transcription Kit with RNase Inhibitor (Thermo Fisher, Waltham, MA, USA). Gene expression assays were performed using AB 7500 Real Time PCR System (Thermo Fisher, Waltham, MA, USA), TaqMan Gene Expression Master Mix (Thermo Fisher, Waltham, MA, USA) and Taqman Gene Expression Assays for human NLRP3 (Hs00366465_m1), CASP-1 (Hs00354836_m1) and OPA-1 (Hs01047018_m1) (Thermo Fisher, Waltham, MA, USA). The 7500 Real Time PCR software was used, and expression values obtained were normalized to the housekeeping gene ACTB (Hs99999903_m1). The relative quantification was conducted relating to the calibrator sample (untreated condition) using the 2^−∆∆Ct^ method (11846609). The amplification efficiency of the assays was established beforehand and found to be comparable. All samples were analyzed in triplicate.

### 4.4. Determination of Cytokines Release

The analyses of cytokines (including interleukin (IL)-1β, IL-2, IL-4, IL-6, IL-8, IL-17, IFN-γ and TNF-α) were performed on supernatant samples collected at different time points (12, 24 and 48 h) after treatments. The simultaneous quantification of the cytokines was obtained using a magnetic bead-based multiplex immunoassays (Bio-Plex^®^, BIO-RAD Laboratories, Hercules, CA, USA). Samples were processed following manufacturer’s instructions and read using the Bio-Plex^®^ 200 reader, while a digital processor managed data output and the Bio-Plex Manager^®^ software (BIO-RAD Laboratories, Hercules, CA, USA) computed data as Median Fluorescence Intensity (MFI) and concentration expressed in pg/mL.

### 4.5. Cellular Metabolism Analysis

The mitochondrial Oxidative Phosphorylation (OXPHOS) pathway was evaluated using the MILLIPLEX^®^ MAP Human OXPHOS Magnetic Bead Panel (Merck, Darmstadt, Germany). This kit, based on the Luminex xMAP^®^ technology, permits the simultaneous detection of NADH-ubiquinone oxidoreductase (Complex I), succinate ubiquinone oxidoreductase (Complex II), ubiquinone cytochrome c oxidoreductase (Complex III), cytochrome c oxidase (Complex IV), ATP synthase (Complex V) and nicotinamide nucleotide transhydrogenase (NNT).

Samples were analyzed with the Bio-Plex^®^ 200 reader and data were computed using Bio-Plex Manager^®^ software (BIO-RAD Laboratories, Hercules, CA, USA), which express data as MFI. The samples preparation and quantification were performed following the manufacturer’s instructions.

### 4.6. Data Analysis

Statistical analysis data were calculated as mean ± standard deviation (SD). Statistical significance was calculated using a one-way or two-way ANOVA and Bonferroni post-hoc test correction for multiple comparison. Analysis was performed using GraphPad Prism software, version 5.0 (GraphPad Software, La Jolla, CA, USA).

## 5. Conclusions

All aspects above should prompt considering MitoQ as an effective adjuvant for the treatment of the autoinflammatory diseases characterized by a cholesterol pathway deregulation that involves mitochondrial homeostasis. This study aimed to highlight the role of the mitochondrial activity in the pathogenesis of these diseases, and further studies are necessary to investigate the balance between inflammasome platform and mitochondrial functions.

## Figures and Tables

**Figure 1 ijms-22-04753-f001:**
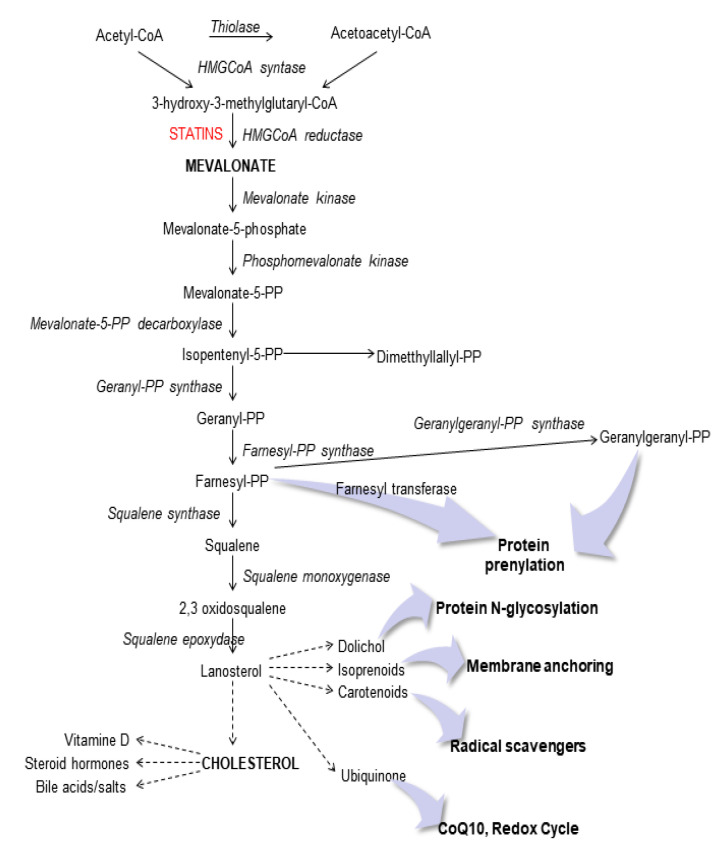
Schematic representation of the mevalonate pathway.

**Figure 2 ijms-22-04753-f002:**
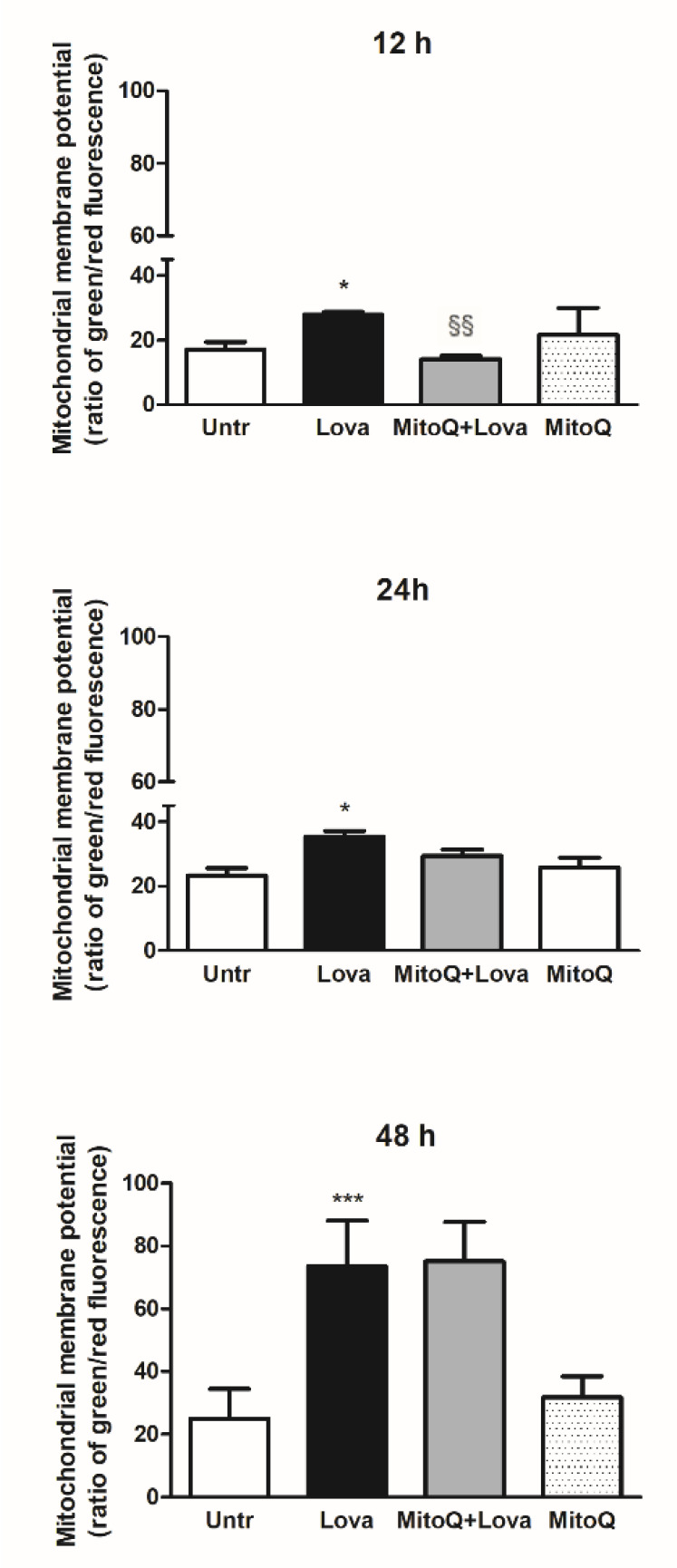
JC-1, a lipophilic cation dye, staining to evidence the mitochondrial membrane potential (*N* = 4 independent experiments). Data are shown as means ± SD (Lova vs. Untreated: * *p* < 0.05; *** *p* < 0.001; MitoQ + Lova vs. Lova: §§ *p* < 0.01 based on one-way ANOVA, post-test Bonferroni). Abbreviations: Untr, untreated; Lova, lovastatin.

**Figure 3 ijms-22-04753-f003:**
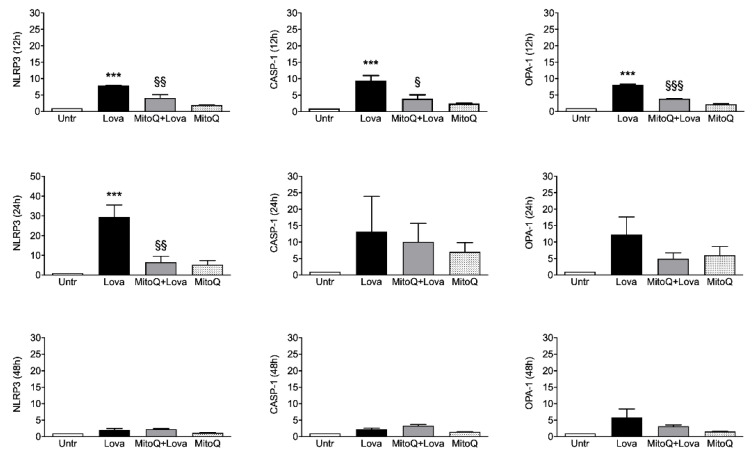
Relative quantification by qPCR of NLRP3, OPA-1 and CASP-1 genes in DAOY cell line (*N* = 4 independent experiments). DAOY cells were incubated for 12, 24 and 48 h with lovastatin and Mitoq and afterward collected for RNA extraction and quantification. Results are reported as mean ± SD. Statistically significant *p*-values are shown (Lova vs. Untreated: *** *p* < 0.001; MitoQ + Lova vs. Lova: § *p* < 0.05; §§ *p* < 0.01; §§§ *p* < 0.001 based on one-way ANOVA, post-test Bonferroni). Abbreviations: Untr, untreated; Lova, lovastatin; NLRP3, NACHT, LRR and PYD domains-containing protein 3; OPA-1, Optic Atrophy 1; CASP-1, caspase 1.

**Figure 4 ijms-22-04753-f004:**
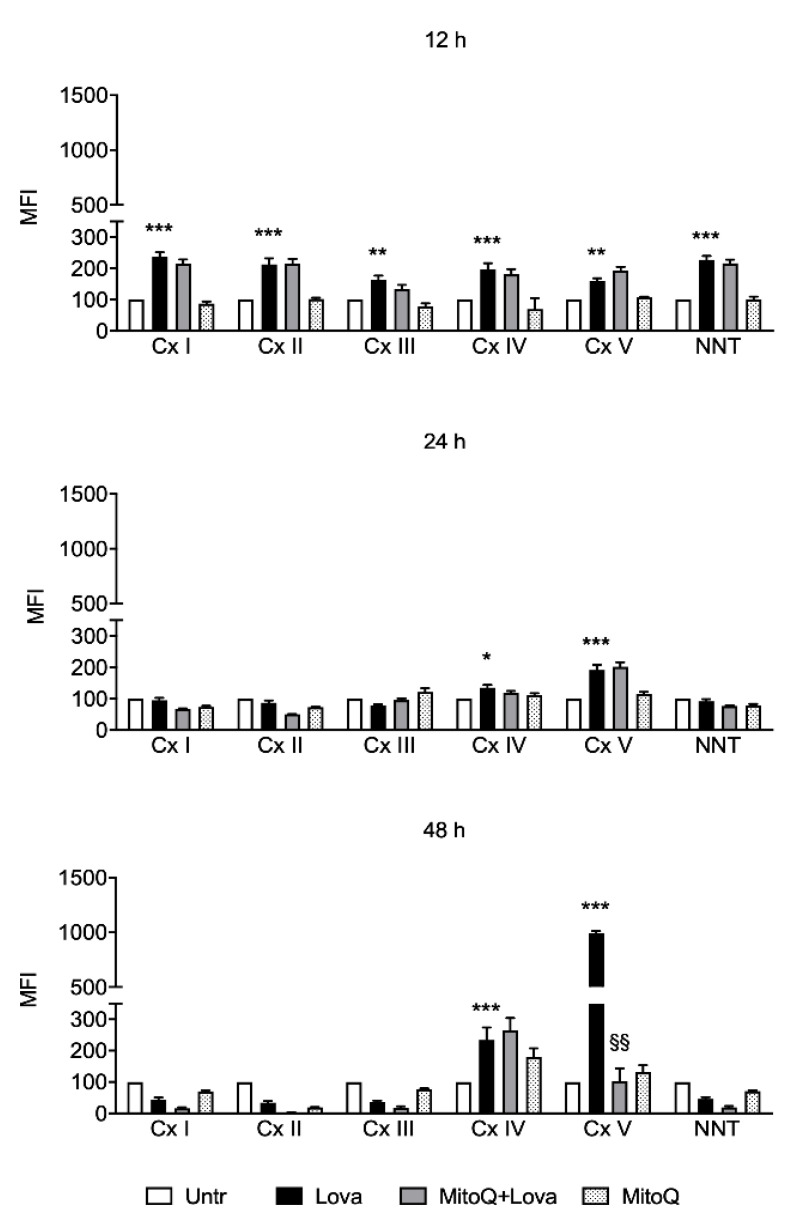
Mitochondrial complexes activity in DAOY cells (*N* = 4 independent experiments). For mitochondrial activity measurement, cells were treated with lovastatin and MitoQ for 12/24/48 h and then analyzed with a MILLIPLEX^®^ human oxidative phosphorylation (OXPHOS) magnetic bead panel. Results are reported as mean ± SD. Statistically significant *p*-values are shown (two-way ANOVA: Lova vs. Untreated: * *p* < 0.05; ** *p* < 0.01; *** *p* < 0.001; MitoQ + Lova vs. Lova: §§ *p* < 0.01). Abbreviations: Untr, untreated; Lova, lovastatin; Cx I, NADH-ubiquinone oxidoreductase (Complex I); Cx II, succinate ubiquinone oxidoreductase (Complex II); Cx III, ubiquinone cytochrome c oxidoreductase (Complex III); Cx IV, cytochrome c oxidase (Complex IV); Cx V, ATP synthase (Complex V); NNT, nicotinamide nucleotide transhydrogenase.

**Table 1 ijms-22-04753-t001:** Cytokine’s quantification at 12 h is expressed in pg/mL. Statistically significant *p*-values are shown (Lova vs. Untreated: * *p* < 0.05; ** *p* < 0.01; MitoQ + Lova vs. Lova: § *p* < 0.05; §§ *p* < 0.01; §§§ *p* < 0.001 based on one-way ANOVA, post-test Bonferroni).

Cytokine (pg/mL)	Untr	Lova	MitoQ + Lova	MitoQ	*p*-Value	
**IL-1β**	10.56 ± 1.46	14.58 ± 0.88	7.75 ± 1.31	12.94 ± 1.73	*	§§§
**IL-2**	41.73 ± 7.07	61.84 ± 1.79	36.72 ± 1.81	48.53 ± 5.03	*	§§
**IL-4**	15.22 ± 1.48	24.02 ± 0.79	13.48 ± 2.48	22.55 ± 4.28	*	§§
**IL-17**	31.70 ± 4.44	50.96 ± 8.42	28.16 ± 7.81	40.25 ± 8.94	*	§
**IL-6**	15.65 × 10^3^ ± 24.56 × 10^2^	59.17 × 10^3^ ± 73.19 × 10^2^	23.34 × 10^3^ ± 15.67 × 10^3^	18.00 × 10^3^ ± 86.42 × 10^2^	**	§
**IL-8**	96.85 × 10^2^ ± 21.37 × 10^2^	22.87 × 10^3^ ± 19.54 × 10^2^	69.60 × 10^2^ ± 23.29 × 10^2^	17.97 × 10^3^ ± 84.44 × 10^2^	*	§
**IFN-γ**	72.24 × 10^1^ ± 20.30 × 10^1^	11.51 × 10^2^ ± 70.48	54.51 × 10^1^ ± 84.15	98.12 × 10^1^ ± 16.65 × 10^1^	*	§§
**TNF-α**	25.75 × 10^1^ ± 39.14	44.59 × 10^1^ ± 29.76	25.33 × 10^1^ ± 60.59	36.44 ×10^1^ ± 69.89	**	§§

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
