# Peer review of "MitoQ Is Able to Modulate Apoptosis and Inflammation"

_ijms, 2021, doi:10.3390/ijms22094753_

Round 1
Reviewer 1 Report
Here, Piscianz et al. investigate the impact of the mitochondrial reactive oxygen species scavenger (mitoQ) on inflammation and cell death in a neuronal cell line exposed to statins, which the authors claim mimics cholesterol de-regulation seen in neurodegenerative and autoimmune disorders. This work provides additional information regarding the mechanism of MitoQ action following disruptions to cholesterol synthesis in DAOY cells, a model previously reported by authors on this manuscript. However, a few key elements of the story remain unexplored. Also, the paper needs considerable work in terms of both content and conclusions. I have several suggestions below:
- The introduction should describe what, if anything, is known about a link between cholesterol, mitochondria, and inflammation. There is a jarring jump to experiments designed to look at inflammatory molecules without a clear rationale.
- A further summary of the prior work on this model is warranted in the introduction. Here the authors state that lovastatin induced mitochondrial dysfunction rather generically without providing details. The authors should, at the very least, mention that lovastatin was cytotoxic though the exact mechanism has not been delineated1.
- Do statins trigger oxidative stress, particularly the accumulation of mitochondrial ROS, in these cells? Simple assays could be used to assess general ROS (e.g. DHE or CM-H2DCFDA dyes) and kits exist for the easy detection of mitochondrial ROS (e.g. MitoSox superoxide indicator). Establishing this first step seems essential to the argument that MitoQ is acting as a ROS scavenger in this model. The timing of such impacts might also provide insights into why the impact of mitoQ is greater in initial stages (e.g. 12 h) and not later (24/48h)
- I’m not sure what the rationale was to split the y-axis for many of the figures (Figure 2) or to use the same scale across all panels in a figure (Figure 3). This somewhat obstructs the ability to see impacts of both lovastatin and mitoQ in the dataset. It is perfectly appropriate to utilize different scales across panels for optimal data visualization.
- The order of data presentation is strange to me. I would think beginning with mitochondrial function (and maybe ROS) then moving to apoptosis and finally inflammation would be the most logical given already published data.
- MitoQ alone appears to induce the release of some cytokines (Supplemental Table) such as IL-6 and INFγ at later time points. The authors fail to note this and do not discuss. Why might this be?
- The discussion needs work.
- The authors need to first summarize the major conclusions of the paper and then move on to addressing the potential importance of these conclusions.
- A model for how exactly mitoQ is acting in the system (either written or visual) is lacking.
- The authors discuss the importance of cholesterol in neurodegeneration but do not make the connection for the reader between neuroinflammation and these diseases. Do patients with Niemann-Pick C or others display a high inflammatory burden? What do we know about inflammation in other diseases such as Alzheimer’s disease?
- What would be the advantages/disadvantages of mitoQ as a therapeutic agent in patients?
- Both the abstract and title do not properly reflect the data contained within this work, nor the ultimate conclusions that can be drawn from the data.
- Title: Nowhere in this work do the authors provide any data relevant to autophagy. I do not know why this is mentioned at all in the title, abstract or introduction. The main conclusions here appear to be that mitoQ modulates apoptosis and inflammation. Should the authors wish to look at autophagy directly they would need to assess impacts on markers such as LC3-II or p62 or look at the formation of LC3-GFP puncta in their cells.
- Abstract:
- Lines 21-23 do not seem to properly reflect the goal of the study. Here the authors assessed the impact of cholesterol de-regulation on inflammasome recruitment and inflammatory cytokine production and the ability of mitoQ to mitigate these impacts.
- Nowhere do the authors measure autophagy (line 27, line 31)
- The authors need to explain what MitoQ did exactly (e.g. decreased mitochondrial permeabilization, cyotokines, apoptosis, etc…) vs labeling this as “a protective effect”
- Final senetence: Re-write, please mention inflammation here and not autophagy
Word choice/grammatical errors
There are several word choice/grammatical errors throughout the manuscript and, in general, it would benefit from an editorial once over by a native English speaker. I’ve highlighted a few of the more obvious errors.
- Line 19: “Several literature data” could be replaced with “Prior studies”
- Line 21: replace “played” with “plays”
- Line 25: Delete “like this”
- Line 39: “have been identified effective in tackling these diseases” is confusing. Perhaps “have proven effective in decreasing oxidative stress across multiple disease models”
- The sentence on lines 40-41 is incomplete (deleting “which” would solve this)
- Line 42: replace “literature data” with “prior studies” or related
- Re-write lines 43-45 to indicate that 1) cholesterol is an essential component of biological membranes. 2) mitochondrial cholesterol is important for mitochondrial biogenesis and membrane maintenance. 3) Mitochondria are dynamic organelles and must be trafficked over long distances to reach synapses in neurons. As a result, neurons are particularly sensitive to disruptions in mitochondrial fusion/fission. 3) Dysregulation in these processes has been linked to neurodegeneration.
- Line 47: “the block” should be “blockade”
- Line 58: “mediates” should be “mediate”
- Line 73-74: This sentence appears to be a little backwards. The goal of the study appears to be to evaluate how altered cholesterol synthesis impacts activation of the inflammasome and the ability of mitochondrial ROS scavengers to mitigate such impacts.
- Line 87-88: Delete “(12 and 24 hours), although it shows an increased trend, also at 12 and 24 hours statistically significative”
- Line 90: Delete “also in this assay,”; change “limited the apoptosis rate” to “limited apoptosis”
- Line 92: Delete “in both timing”
- Line 139-141: I don’t understand this sentence at all. The data indicate that lovastatin elevates activity across complexes initially, and only complex IV and V at later time points. MitoQ only reverses these effects at the 48 hour timepoint.
- Throughout: Replace the use of “timing” to refer to “timepoints”
- Line 153: “indeed is about 10-fold major than in any other organ” should be “containing about 10-fold more than any other organ”
- Line 157: Replace “This property is most notably in the several disorders associated with the defect in the cholesterol synthesis and metabolism” with “Notably, several disorders are associated with defects in cholesterol synthesis and metabolism”
- Line 159: delete “and”
- Line 160: insert “and” before “Smith”
- Line 163: Delete “the” before “inflammation”
- Line 169: Replace “in a long time” with “at later timepoints”
- Line 180: Replace “unlike” with “in contrast”
References
- 1 Marcuzzi, A. et al. Neuronal Dysfunction Associated with Cholesterol Deregulation. Int J Mol Sci 19, doi:10.3390/ijms19051523 (2018).
Reviewer 2 Report
6 April 2021
Review on the manuscript titled “MitoQ plays a pleiotropic role in the autophagy mechanism in neuroinflammation” by Piscianz E, submitted to International Journal of Molecular Sciences.
Dear Authors,
The mitoquinone (MitoQ) is a reactive oxygen species scavenger known to be neuroprotective. The authors examined the effects of MitoQ on the neuronal cell line (DAOY) treated with statin which blocks the mevalonate pathway and studied the expression of the genes involved in autophagy and mitochondrial activities, cytokine expressions, and mitochondrial complexes. MitoQ showed a protective effect against statin in 12 hours of time, which did not last for 24 or 48 hours. The authors concluded MitoQ as an effective adjuvant for the treatment of the inflammatory disease involved in the cholesterol metabolism of mitochondria.
Please reconsider the following parts:
- Pages 1, Lines 30-32: Conclusion is not so relevant to the result in the text. Please refine the conclusion.
- Page 2, Lines 55-58: Please rephrase the fist sentence of the paragraph in connection with the previous paragraph. Cholesterol metabolism -> inflammation.
- Pages 2, 3, Introduction: Please clarify the relationship between cholesterol metabolism, the mevalonate pathway, inflammation, autophagy, and mitochondria function.
- Page 4, Figure 2 caption: Please define the abbreviations presented in the figure.
- Page 5, Figure 3 caption: Please define the abbreviations presented in the figure.
- Page 6, Figure 4 caption: Please define the abbreviations presented in the figure.
- Pages 8, 9, Conclusion: Please expand the conclusion.
- Pages 7-10, references: Please cite more references. The number of references is preferable at least more than 50 for research article.
The manuscript contains four figures, one table, and 30 references. The manuscript carries important value presenting potential beneficial effects of MitoQ for mitochondrial diseases. However, the concepts of and the relationships between neurodegenerative diseases, autoinflammatory diseases, reactive oxygen species, cholesterol metabolism, the mevalonate pathway, mitochondrial homeostasis and dysfunction, autophagy, and genetic mitochondrial diseases were not clearly presented. Thus, it makes obscure the purpose of the study. It also deserves to clearly present the rationale of each assay and the choice of cytokines and the gene expression, the significance of the results of the mitochondrial complex study and finally significance of the results overall. I reconsider this manuscript for publication after major revision.
Reviewer 3 Report
The intent of the study was to examine the ability of MitoQ to protect cells from a statin. Statins reduce cholesterol synthesis and the authors previously published that this causes ROS and cell death. There are several concerns about the study:
- MitoQ is not a drug used to treat any diseases in humans currently but its stated in the introduction that several mitochondrial targeted antioxidants are used to treat diseases in humans. This drug is not FDA approved although maybe its approved for use in other countries outside the USA (this should be clarified).
- JC1 is not a dye that can distinguish healthy from apoptotic cells. JC1 when examining the ratio of red and green fluoresence is reporting on mitochondrial membrane potential. It should be used with a counterstain to normalize to cell number (like DAPI or Hoechst). Apoptosis should be examined using an appropriate dye like Annexin V, trypan blue, etc.
- Its unclear how examining the expression of mitochondrial complexes reflects their activity levels. Typically these two measures are not equivalent. Expression changes do not equal activity changes.
- The conclusions do not discuss the effects of MitoQ alone which are typically observed especially on JC1 green intensity and cytokine production. In some cases it appears MitoQ increases cytokine expression similar to the statin.
- ROS was not measured and this seems to be an important factor in the hypothesis/conclusion.
- Overall the discussion is confusing especially since mitochondrial function was not measured.
Round 2
Reviewer 1 Report
The article is much improved. I think the data warrant publication but have a few minor concerns regarding the overall structure of the paper that should be addressed prior to publication:
- The abstract is redundant and contains mostly background information.
- The background provided can be streamlined
- This section “The present study aimed to examine the impact of the inflammation platform activation on the neuronal cell line (DAOY) treated with specific inflammatory stimuli and whether MitoQ addition can modulate these deregulations. DAOY cells were pre-treated with MitoQ and then stimulated by a blockade of the cholesterol pathway, also called mevalonate pathway, using a statin, mimicking the cholesterol deregulation, a common parameter presents in some neurodegenerative and autoinflammatory diseases” can be shortened: “Here, we investigated the impact of MitoQ on inflammation in the neuronal cell line DAOY following blockade of cholesterol synthesis with a statin, which mimics the cholesterol deregulation observed in neurodegenerative and autoinflammatory diseases.”
- The authors need to include a brief summary of key conclusions (replacing the sentences on lines 26-30:
- MitoQ ameliorates the impact of statins on mitochondrial membrane potential
- MitoQ decreases up-regulation of key inflammasome components
- MtioQ reduces inflammatory cytokine production
- Statin-dependent changes in expression of ETC components is partially mitigated with MitoQ
- The background provided can be streamlined
- Line 114: delete “which induces the activation of inflammation”, redundant
- Line 129-131. The section “The present study aimed to identify the components of the inflammation platform and to examine the impact of this inflammatory system on the mitochondria functions, associated with the blockade of mevalonate pathway, in a human derived desmoplastic cerebellar medulloblastoma cell line used as a neuronal model” is confusing and could be simplified to something like “The present study aimed to asses the impact of mitoQ on mitochondrial function and inflammation following blockade of the mevalonate pathway in human derived desmoplastic cerebellar medulloblastoma cells (DAOY), a neuronal model”
- Line 395: Replace “condition in which MitoQ action is not effective” with “, which MitoQ fails to effectively mitigate”
- Delete the sentence spanning lines 395-397, it is unnecessary
- Line 401: Replace “the morphologic change in a short time” with “mitochondrial integrity in the acute phase following disruption in cholesterol biosynthesis” or similar
- Line 402: Replace “It is important, instead, to highlight that the high significant OPA-1 expression level after 12 hours of lovastatin treatment is strictly associated with its” with something like “Indeed, OPA-1 expression is significantly elevated at 12 hours and modulated by MitoQ emphasizing the importance of mitochondrial morphology at this stage due to OPA-1’s…”
- Line 426: Replace “valuable” with “visible”
Reviewer 2 Report
22 April 2021
The 2nd review on the manuscript titled “MitoQ plays a pleiotropic role in the autophagy mechanism in neuroinflammation” by Piscianz E, submitted to International Journal of Molecular Sciences.
Dear Authors,
The mitoquinone (MitoQ) is a reactive oxygen species scavenger known to be neuroprotective. The authors examined the effects of MitoQ on the neuronal cell line (DAOY) treated with statin which blocks the mevalonate pathway and studied the expression of the genes involved in autophagy and mitochondrial activities, cytokine expressions, and mitochondrial complexes. MitoQ showed a protective effect against statin in 12 hours of time, which did not last for 24 or 48 hours. The authors concluded MitoQ as an effective adjuvant for the treatment of the inflammatory disease involved in the cholesterol metabolism of mitochondria.
The manuscript contains four figures, one table, and 56 references. The revised manuscript was improved substantially. The conclusion is expected to expand more to contain short summaries of the study, results, interpretation, and future perspective. The interpretation should include the significance of the results of the mitochondrial complex study and the results overall. The future perspective should include the concrete descriptions of authors’ view on future directions. The manuscript carries important value presenting potential beneficial effects of MitoQ for mitochondrial diseases I recommend this manuscript for publication after minor revision.
I declare no conflict of interest regarding this manuscript.
Author Response
The 2nd review on the manuscript titled “MitoQ plays a pleiotropic role in the autophagy mechanism in neuroinflammation” by Piscianz E, submitted to International Journal of Molecular Sciences.
Dear Authors,
The mitoquinone (MitoQ) is a reactive oxygen species scavenger known to be neuroprotective. The authors examined the effects of MitoQ on the neuronal cell line (DAOY) treated with statin which blocks the mevalonate pathway and studied the expression of the genes involved in autophagy and mitochondrial activities, cytokine expressions, and mitochondrial complexes. MitoQ showed a protective effect against statin in 12 hours of time, which did not last for 24 or 48 hours. The authors concluded MitoQ as an effective adjuvant for the treatment of the inflammatory disease involved in the cholesterol metabolism of mitochondria.
The manuscript contains four figures, one table, and 56 references. The revised manuscript was improved substantially. The conclusion is expected to expand more to contain short summaries of the study, results, interpretation, and future perspective. The interpretation should include the significance of the results of the mitochondrial complex study and the results overall. The future perspective should include the concrete descriptions of authors’ view on future directions. The manuscript carries important value presenting potential beneficial effects of MitoQ for mitochondrial diseases I recommend this manuscript for publication after minor revision.
I declare no conflict of interest regarding this manuscript.
Authors’ reply: According to reviewer suggestion, we rewrote part of Conclusions to improve the quality of scientific description.
Reviewer 3 Report
The authors have addressed all comments.
Author Response
REVIEWER 3:
The authors have addressed all comments.
Authors’ reply: We thank Reviewer 3 for the positive comments.